# Assimilation of wind data from airborne Doppler cloud-profiling radar in a kilometre-scale NWP system

Mary Borderies[1], Olivier Caumont[1], Julien Delanoë[2], Véronique Ducrocq[1], and Nadia Fourrié[1]

[1]CNRM UMR 3589, Université de Toulouse, Météo-France, CNRS, Toulouse, France
[2]LATMOS, IPSL, Université Versailles St-Quentin, CNRS, UPMC, Guyancourt, France

**Correspondence:** mary.borderies@meteo.fr

**Abstract.** The article reports on the impact of the assimilation of wind vertical profile data in a kilometre-scale NWP system on predicting heavy precipitation events in the north-western Mediterranean area. The data collected in diverse conditions by the airborne W-band radar RASTA (Radar Airborne System Tool for Atmosphere) during a 45-day period are assimilated in the 3-h 3DVar assimilation system of AROME. The impact of the length of the assimilation window is investigated. The data assimilation experiments are performed for a heavy rainfall event, which occurred over south-eastern France on 26 September 2012 (IOP7a), and over a 45-day cycled period. Results indicate that the quality of the rainfall accumulation forecasts increases with the length of the assimilation window, which recommends to use observations with a large period centred on the assimilation time. The positive impact of the assimilation of RASTA wind data is particularly evidenced for the IOP7a case since results indicate an improvement in the predicted wind at short term ranges (two and three hours) and in the 11-h precipitation forecasts. However, on the 45-day cycled period, the comparison against other assimilated observations shows an overall neutral impact. Results are still encouraging since a slight positive improvement in the 5-, 8- and 11-h precipitation forecasts was demonstrated.

## 1 Introduction

The Mediterranean area is frequently subject to heavy precipitation events, causing heavy damages and human losses (Ducrocq et al., 2014). Over the last years, Numerical Weather Prediction (NWP) models have been operationally implemented to improve the accuracy and timely prediction of such severe weather. The quality of the predictions depends, among others, on the initial atmospheric state. Several studies suggested that the impact of the assimilation of wind observations was beneficial on analyses and forecasts (Horányi et al., 2015).

Over land, ground-based Doppler precipitation radar data are now operationally assimilated in kilometre-scale NWP systems since their potential to improve the short-term forecasts has been demonstrated (Montmerle and Faccani, 2009; Simonin et al., 2014). In clear air condition, wind observations can be provided by insect-derived Doppler radar measurements (Kawabata et al., 2007; Rennie et al., 2011) or by Doppler lidars (Weissmann et al., 2012; Kawabata et al., 2014). To fill the gap in clear air conditions, radar wind profilers provide vertical profiles of the horizontal wind at a high vertical resolution. Several studies

highlighted the benefit of the assimilation of these data into NWP models to improve short-term forecasts (Benjamin et al., 2004; Illingworth et al., 2015b). However, the main drawback of ground-based radars and radar profilers is that they are only distributed over land.

Because wind observations are too sparse over ocean, Atmospheric Motion Vectors (AMV) are now operationally derived using the movement of cloud and water vapour tracers from consecutive satellite images. They provide tropospheric wind data measurements at a global scale with a high temporal resolution. Recent studies indicate an overall positive impact of the assimilation of AMV data in NWP models on the subsequent forecasts (Deb et al., 2016; Kumar et al., 2017). Nevertheless, contrary to most active sensors, AMV measurements do not provide wind vertical profile informations but only cloud top informations.
Besides, there is an uncertainty in the knowledge of the observed cloud top level (Salonen et al., 2015).

To fill the gap within the existing observing system, Baker et al. (2014) highlighted the need for extra wind vertical profile measurements over ocean to improve the initial conditions for NWP forecasts. This need for new wind measurements particularly applies in the Mediterranean region since offshore convective systems, which are responsible for heavy precipitation
events, are not well predicted by kilometre-scale NWP models (Duffourg et al., 2016; Martinet et al., 2017). In the near future, the Doppler W-band radar on-board the EarthCare satellite mission (scheduled to be launched in middle 2021, Illingworth et al., 2015a) will provide for the first time vertical profiles of wind data from Doppler radar at a high vertical resolution over land and over sea. In the meantime, the WIVERN satellite concept mission carrying a conically scanning Doppler W-band radar is also being conceived (Illingworth et al., 2018). So far, the impact of the assimilation of wind vertical profiles from
W-band radar has never been investigated.

Airborne Doppler radars have the advantage of collecting a large dataset of measurements over land and sea at very fine scales. Pu et al. (2009) showed that the 3DVar assimilation of wind data from airborne Doppler radar results in significant improvement in the intensity and precipitation forecasts of Hurricane Dennis. Following on, Li et al. (2014) demonstrated the
benefits of the 4DVar assimilation of the ELDORA X-band radar velocity data in the simulation of Hurricane Nuri's genesis. The positive impact due to airborne Doppler velocity data assimilation for hurricane forecasts has also been investigated with an Ensemble Kalman Filter by Weng and Zhang (2012). So far, this kind of study has never been done in the Mediterranean area. In addition, the measurements used in the hurricane studies listed above were collected with side-looking radar (elevation angle $\leq 70°$) at lower frequencies (X or C bands).

The primary objective of this article is to evaluate for the first time the impact of assimilating wind profiles retrieved by airborne W-band radar in a kilometre-scale NWP model. The current study covers a two-month period with the airborne Doppler W-band radar RASTA (Radar Airborne System Tool for Atmosphere) during the HyMeX (HYdrological cycle in the Mediterranean EXperiment, Drobinski et al., 2014) first Special Observing Period (HyMeX-SOP1, Ducrocq et al., 2014) over a region
of the Mediterranean area prone to heavy rainfall. The main goal of the HyMeX-SOP1 was to document the heavy precipita-

tion events that regularly affect northwestern Mediterranean coastal areas. RASTA is a multi-beam antenna system (6 beams in total) that allows the documentation of the three components of the wind field in the vertical at a high resolution of 60 m and quasi-continuously in time during the flights. The current assimilation study is performed in a quasi-operational framework, using a version of the Météo-France operational kilometre-scale model AROME (named AROME-WMed) specifically designed for the HyMeX-SOP1, with its 3DVar assimilation system associated with a 3-h assimilation cycle.

To assess the potential of RASTA wind data to improve short-term forecasts, a series of experiments are first conducted for a heavy rainfall event, which occurred during the Intensive Observation Period 7a (IOP7a) over south-eastern France on 26 September 2012. Next, a cycling data assimilation run is conducted over a 45-day period from 24 September 2012 to 5 November 2012 in order to study the impact of the assimilation of RASTA wind data in various conditions during the whole HyMeX-SOP1. This article investigates the impact of the choice of the assimilation window in a 3DVar system. Indeed, data from moving platforms, such as RASTA, have the disadvantage of not being measured simultaneously at the assimilation time, but over the flight leg. A small assimilation window constrains the number of assimilated data to those who are nearly valid at the assimilation time. By contrast, a larger assimilation window leads to larger coverage, but with observations which might be no longer valid. Therefore, a sensitivity study to the assimilation window is performed in this study.

This article is organized as follows. In section 2, the airborne Doppler W-band radar RASTA and the period of study are described. The kilometre-scale NWP model AROME-WMed with its 3DVar assimilation system are then presented in section 3. Following on, the different model simulations are detailed in section 4. Finally, the evaluation of the different experiments is first focused on IOP7a in section 5, followed by a statistical evaluation over the whole HyMeX SOP1 in section 6.

## 2 Radar data and period of study

The Doppler W-band radar RASTA is first described in Section 2.1, and details about the data collected by RASTA during the HyMeX first Special Observing Period (SOP1) field campaign are then briefly given in Section 2.2.

### 2.1 The Doppler W-band radar RASTA

The airborne cloud radar RASTA is a monostatic Doppler multi-beam antenna system operating at 95 GHz (Bouniol et al., 2008, Protat et al., 2009, Delanoë et al., 2013). The aircraft platform used is the French Falcon 20 research aircraft from the SAFIRE unit (Service des Avions Français Instrumentés pour la Recherche en Environnement). This unique instrument allows the documentation of the microphysical properties and the three components of the wind field in the vertical at a high resolution of 60 m and quasi-continuously in time during the flights.

The radar RASTA is equipped with six Cassegrain antennas pointing either upward (antennas 1-3) or downward (antennas 4-6). Therefore, RASTA measures the reflectivity and the radial velocity in three non-collinear directions above and below

the aircraft in the clouds. A schematic figure of RASTA configuration during the HyMeX-SOP1 is given by Bousquet et al. (2016), their Figure 1. The radial velocity measurements are collected at a vertical resolution of 60 m and a time resolution of 250 ms (i.e. 1.5 s between two measurements of the same antenna). The maximum range is 15 km with a Nyquist velocity of 7.8 m s$^{-1}$ (the Pulse Repetition Frequency equals 10 kHz).

The data processing described by Bousquet et al. (2016) is applied to RASTA wind observations. First, the exact speed of the aircraft and the pointing angles are used to rigorously determine the component related to the aircraft's movement. Doppler measurements are then processed by removing the projection of aircraft ground speed along the six antenna beams. Next, Doppler velocities are unfolded using in situ wind sensor for the first gate and by applying a gate to gate correction for the other gates. In addition to that the combination of the three non-collinear beams is used to verify potential unfolding issues as the retrieval would be locally inconsistent. For ground-pointing antennas, a check-up is conducted in order to ensure that ground return velocities are close to 0 m s$^{-1}$. Upward-looking antennas errors are estimated and corrected by ensuring continuity between the data collected above and below the aircraft. After processing, the Doppler velocity of the three downward-looking and upward-looking antennas are combined to retrieve the horizontal and vertical wind components above and below the aircraft. More details on the RASTA configuration during HyMeX can be found by Bousquet et al. (2016). The retrieved horizontal wind components will be assimilated in the 3DVar assimilation system of AROME-WMed.

## 2.2  RASTA data during the HyMeX first Special Observing Period (SOP1)

This study takes advantage of the data collected by RASTA during the HyMeX SOP1, which took place from 5 September to 5 November 2012 over the western Mediterranean (Ducrocq et al., 2014). The main goal of the SOP1 was to document the heavy rainfall events that regularly affect northwestern Mediterranean coastal areas. During the two-month campaign, approximately 20 rainfall events were documented in France, Italy and Spain (Ducrocq et al., 2014). Specifically, the RASTA radar aboard the Falcon 20 collected data during 18 flights in and around mesoscale convective systems in diverse conditions.

The data collected by RASTA during the SOP1 offer a wide variety of conditions over land, sea and complex terrains. Among all the observed vertical columns over the SOP1, 72.6% were collected in stratiform areas and 13.1% in clear sky and 14.3% in convective areas (Borderies et al., 2018). RASTA flight paths during the HyMeX SOP1 are represented by the black lines in Figure 1.

## 3 Model and data assimilation system

### 3.1 The AROME-WMed NWP model

This study is conducted with AROME-WMed (Fourrié et al., 2015), the HyMeX-dedicated version of the Météo-France operational kilometre-scale NWP model AROME (Seity et al., 2011). AROME-WMed, which covers the entire northwestern Mediterranean Basin, was specially designed for the HyMeX-SOP1 and ran in real time to plan the airborne operations in advance, especially in the mesoscale convective systems. AROME-WMed is based on the AROME-France version operationally employed in 2012: the deep convection is explicitly resolved and the microphysical processes are governed by the ICE3 one-moment bulk microphysical scheme (Pinty and Jabouille, 1998). AROME-WMed runs at a horizontal resolution of $2.5 \times 2.5$ km with 60 vertical levels ranging from approximately 10 m above ground level to 1 hPa. Compared to AROME, AROME-WMed covers an extended domain centred on the northwestern Mediterranean area. The AROME-WMed domain is displayed in Figure 1. It has $948 \times 628$ horizontal grid points, which is equivalent to a horizontal size of $2370 \times 1570$ km$^2$. In addition, to increase the observation coverage in the southern part of the domain, more satellite (AMSU) and ground-based Spanish weather station observations are assimilated in AROME-WMed.

### 3.2 3DVar assimilation system

AROME-WMed has a three-dimensional variational (3DVar) data assimilation system (Brousseau et al., 2011) associated with a 3-h assimilation cycle. It is based on an incremental formulation (Fischer et al., 2005) and the control variables are temperature, specific humidity, surface pressure, vorticity and divergence. AROME-WMed background error covariances were computed using a period in October 2010 characterized by convective systems over the northwestern Mediterranean region (Fourrié et al., 2015).

Every 3 hours an analysis is computed by using all observations available within a $\pm$ 1 h 30 min assimilation window and a 3-h forecast to produce a first guess for the next cycle. The assimilation system ingests a wide variety of observations from satellite, ground-based Global Navigation Satellite System (GNSS), aircraft, radiosondes, drifting buoys, balloons and wind profilers, automatic land and ship weather stations, and ground-based radars of the French network ARAMIS (reflectivity and radial velocity).

## 4 Data assimilation experiments

To assess the potential of RASTA wind data to improve short-term forecasts of heavy precipitation events, a total of 4 experiments is conducted over a 45-day cycled period during the HyMeX-SOP1. Focus is also made on one of the most significant episodes which occurred within France during the HyMeX SOP1 campaign on 26 September 2012.

## 4.1  RASTA wind data pre-processing

First, "super-observations" are created to reduce observation and representativeness errors. They are calculated by interpolating RASTA wind data in the model vertical and horizontal resolutions. This interpolation is done by taking the median value of all data available along the aircraft track within a box of 2.5 km length between the two half model levels surrounding each
model level. Applying a median filter instead of averaging allows to reduce the influence of outliers, due to the difficulty of having high quality measurements for airborne Doppler radar (Bosart et al., 2002). Indeed, after the data processing described in subsection 2.1, some spurious data were still occasionally present. Using a median filter, instead of a mean filter, helps to reduce the weight that these spurious observations can have in the calculation of RASTA wind "super-observations".

When the aircraft roll and/or pitch angles are too high (ie., if $d = \sin(\theta) \times R \geq \frac{2.5}{2}$ in Figure 2, with R the range from the
radar), some data might not be in the same box at a given range from the aircraft (for instance in the box number 3 in Figure 2). Therefore, these data are not taken into account.

After this pre-processing, to satisfy assumptions about observation error covariances, which are supposed to be 0 $m^2 \ s^{-2}$, a thinning is applied to RASTA wind "super-observations". One super-observation out of three is then assimilated, which is equivalent to approximately one observation every 5 km to 9 km depending on the aircraft speed.

## 4.2  Experimental setup

RASTA wind data are not measured simultaneously, but over the flight leg. Therefore, at each assimilation time $T$ from 00 UTC to 21 UTC, the 3DVar assimilation system of AROME-WMed ingests all RASTA wind data available during an assimilation window $\Delta_t$ centred on the assimilation time $T$, as if they were valid at the time $T$. Too large an assimilation window $\Delta_t$ would result in assimilating data that are no longer valid at the current assimilation time $T$, especially for convective systems which
can evolve quickly in time. On the other hand, it is likely that the impact will be neutral if the assimilation window is too short, because less data are assimilated. Therefore, the impact of the assimilation of RASTA wind data is tested with three different assimilation windows $\Delta_t$: 3 hours (RASTA$_{3\,h}$), 2 hours (RASTA$_{2\,h}$) and 1 hour (RASTA$_{1\,h}$) centred on the assimilation time $T$.

Finally, four different experimental designs are defined. The analyses of the global operational NWP model ARPEGE are
used to initialise the experiments and to provide boundary conditions. In the control (CTRL) experimental design, only the observations that are operationally assimilated are taken into account (see subsection 3.2). The three additional RASTA experimental designs (RASTA$_{3\,h}$, RASTA$_{2\,h}$ and RASTA$_{1\,h}$) share the same configuration as CTRL, but include the assimilation of RASTA wind data every 3-h from 00 UTC to 21 UTC.

Because the Doppler multi-beam antenna system of RASTA can retrieve the horizontal wind components ($u$, $v$), which are linked to two control variables of AROME-WMed (vorticity and divergence), the assimilation of RASTA wind data is straightforward and does not require the use of a radial wind observation operator. *Bousquet et al. (2016) demonstrated that the root-mean-square error of RASTA wind data versus ground-based centimetre-wavelength radars is on the order of 4 m*

$s^{-1}$. *They argued that this error mainly originated from the sampling volume of ground-based radars being much larger than that of RASTA.* In this study, it has been decided to use the same observation error as the one used for radiosondes, which increases with the altitude (from $\approx 1.8~\mathrm{ms}^{-1}$ at 900 hPa to $\approx 2.52~\mathrm{ms}^{-1}$ at 200 hPa). Finally, in addition to the pre-processing described in subsection 4.1, a quality control is also performed prior to the assimilation: observations with innovation (Obser-vations - Background) greater than a threshold are rejected. This threshold depends on both the observation and background errors.

First, the four different experimental designs are run during a 45-day cycled period from 00 UTC 24 September 2012, which is the day when the Falcon 20 first flew during HyMeX-SOP1, to 5 November 2012, after the last flight. During this period, the different assimilation experiments are named $\mathrm{CTRL}^{\mathrm{SOP1}}$, $\mathrm{RASTA}^{\mathrm{SOP1}}_{3\,\mathrm{h}}$, $\mathrm{RASTA}^{\mathrm{SOP1}}_{2\,\mathrm{h}}$ and $\mathrm{RASTA}^{\mathrm{SOP1}}_{1\,\mathrm{h}}$. The number of assimilated data over the covered period is represented as a function of the pressure level in Figure 3 for the three RASTA experiments. Table 1 summarizes the different assimilation experiments. The fourth column shows the percentage of analyses in which RASTA wind data were assimilated over the total number of analyses (360) during the 45-day cycled period for the different RASTA experiments. A larger assimilation window results in assimilating data more frequently, but the time lag between the observation time and the analysis time is greater than one hour. On the other hand, a smaller assimilation window constrains the number of analyses to those for which the observations are valid near the analysis time. Therefore, the percentage of analyses in which RASTA wind data were assimilated decreases with the length of the assimilation window from 9.5% in the $\mathrm{RASTA}^{\mathrm{SOP1}}_{3\,\mathrm{h}}$ experiment to 7.2% in the $\mathrm{RASTA}^{\mathrm{SOP1}}_{1\,\mathrm{h}}$ experiment. Finally, the last column of Table 1 represents the percentage of RASTA wind data which were assimilated among the total number of assimilated data (conventional, GNSS, radar, satellite, RASTA, etc.) over the entire AROME-WMed domain (represented in Figure 1). This percentage is quite small because of the already dense observing network used in AROME-WMed.

Finally, the four different experimental designs are also run on a heavy precipitation event which occurred during the Intensive Observation Period 7a (IOP7a) on 26 September 2012 during the morning. The $\mathrm{CTRL}^{\mathrm{IOP7}}$, $\mathrm{RASTA}^{\mathrm{IOP7}}_{3\,\mathrm{h}}$, $\mathrm{RASTA}^{\mathrm{IOP7}}_{2\,\mathrm{h}}$ and $\mathrm{RASTA}^{\mathrm{IOP7}}_{1\,\mathrm{h}}$ experiments start at 00 UTC 26 September 2012 and end at 12 UTC 26 September 2012.

**Table 1.** Experimental design from 24/09/2012 to 05/11/2012

| Experiment | Assimilated data | $\Delta_t$ | RASTA analyses | Percentage of assimilated RASTA data |
|---|---|---|---|---|
| $\mathrm{CTRL}^{\mathrm{SOP1}}$ | Conv. + GNSS + radar + satellite | - | 0 | 0% |
| $\mathrm{RASTA}^{\mathrm{SOP1}}_{3\,\mathrm{h}}$ | $\mathrm{CTRL}^{\mathrm{SOP1}}$ + RASTA | 3h | 9.5% (35 cases out of 360) | 4.55% |
| $\mathrm{RASTA}^{\mathrm{SOP1}}_{2\,\mathrm{h}}$ | $\mathrm{CTRL}^{\mathrm{SOP1}}$ + RASTA | 2h | 8.9% (32 cases out of 360) | 3.34% |
| $\mathrm{RASTA}^{\mathrm{SOP1}}_{1\,\mathrm{h}}$ | $\mathrm{CTRL}^{\mathrm{SOP1}}$ + RASTA | 1h | 7.2% (26 cases out of 360) | 1.9% |

## 5 Results on the case study

The impact of RASTA wind data is first illustrated on a heavy precipitation event which occurred during the Intensive Observation Period 7a (IOP7a) on 26 September 2012.

### 5.1 Case description: IOP7a

The IOP7a precipitation event is one of the most significant episodes that occurred within France during the HyMeX SOP1 campaign (Hally et al., 2014). This case study was located over south-eastern France in the area delimited by the red box in Figure 1, which has been enlarged in Figure 4. The precipitation event consisted in a convective line over the mountainous region and a band of stratiform rainfall over the Gard and the Ardèche departments. More than 100 mm of rain were observed between 00:00 UTC on 26 September and 00:00 UTC on 27 September. A first peak of rainfall accumulation is observed in the morning at 08:00 UTC and a second one in the late afternoon at 17:00 UTC. This event is further described by Hally et al. (2014).

During the IOP7a, RASTA data were collected during Flight 15 between 06:10 and 09:45 UTC. Therefore, RASTA wind data are assimilated for the first time at 06:00 UTC. Since the Falcon 20 took off at 06:10 UTC, the $RASTA_{1h}^{IOP7}$ experiment assimilates all the RASTA wind data that are available between 06:10 UTC and 06:30 UTC, as if they were valid at 06:00 UTC. Similarly, the $RASTA_{2h}^{IOP7}$ ($RASTA_{3h}^{IOP7}$) experiment assimilates RASTA wind data until 07:00 UTC (07:30 UTC) as if they were valid at 06:00 UTC.

The observation time along the aircraft flight path is represented by the colour data points in Figure 4. Figure 4 shows that data were mainly collected in the area where the band of rainfall was located, over the Ardèche and the Gard departements. In particular, most of the data that are assimilated at the 06:00 UTC analysis (before an observation time of 07:30 UTC) are located upwind of where the rainfall event occurred over the Ardèche department. Therefore, the assimilation of RASTA wind data at 06:00 UTC is expected to have an impact on the forecasts, especially for the first peak of rainfall accumulation which occurred in the morning.

### 5.2 Impact on analyses

Figure 5 shows (from the top to the bottom) the wind speed (left panels, A to E) and the wind direction (right panels, F to J) for the observations, the $CTRL^{IOP7}$, the $RASTA_{1h}^{IOP7}$, the $RASTA_{2h}^{IOP7}$ and the $RASTA_{3h}^{IOP7}$ analyses. The different analyses were computed using the same background state. The three different assimilation windows $\Delta_t$ are delimited by the vertical lines.

As expected, Figure 5 indicate a better agreement with the observations if RASTA wind data are assimilated, in terms of both direction and speed. The $RASTA_{3h}^{IOP7}$, $RASTA_{2h}^{IOP7}$ and $RASTA_{1h}^{IOP7}$ experiments assimilate all the observations until

06:30 UTC, 07:00 UTC and 07:30 UTC, respectively. These different time limitations explain the differences in wind and direction between the different RASTA experiments.

Even though the three RASTA analyses are very similar to each other within their respective assimilation windows $\Delta_t$, at 06:30 UTC the $\text{RASTA}_{3\,\text{h}}^{\text{IOP7}}$ (panel E) and the $\text{RASTA}_{2\,\text{h}}^{\text{IOP7}}$ (panel D) experiments exhibit larger velocities at 10 km of altitude than the $\text{RASTA}_{1\,\text{h}}^{\text{IOP7}}$ (panel C) experiment. This discrepancy is explained by the fact that the aircraft does not have a rectilinear trajectory and passes over the same location at several times. In particular, Figure 4 shows that RASTA collected data at the same location at 06:30 UTC and at 07:00 UTC. In such a case, all data are assimilated as if they were valid equally at the assimilation time $T$ (06 UTC here). This overpass explains why the $\text{RASTA}_{3\,\text{h}}^{\text{IOP7}}$ and the $\text{RASTA}_{2\,\text{h}}^{\text{IOP7}}$ are slightly different from the $\text{RASTA}_{1\,\text{h}}^{\text{IOP7}}$ experiment at 06:30 UTC, in terms of both direction and speed. Similarly, there is an overpass of the aircraft at 06:15 UTC and at 07:20 UTC. At this location, while the $\text{RASTA}_{2\,\text{h}}^{\text{IOP7}}$ and the $\text{RASTA}_{1\,\text{h}}^{\text{IOP7}}$ experiments only assimilate the data available at 06:15 UTC, the $\text{RASTA}_{3\,\text{h}}^{\text{IOP7}}$ experiment also assimilates the data collected at 07:20 UTC. However, the wind has increased between 06:00 UTC and 07:30 UTC. Hence, the $\text{RASTA}_{3\,\text{h}}^{\text{IOP7}}$ experiment exhibits at 06:15 UTC higher velocity and different direction (panels E and J) at approximately 10 km of altitude, compared to the $\text{RASTA}_{1\,\text{h}}^{\text{IOP7}}$ (panels C and H) and the $\text{RASTA}_{2\,\text{h}}^{\text{IOP7}}$ (panels D and I) experiments.

Figure 6A represents the wind speed increments at approximately 4 km of altitude (model level 30) between the $\text{RASTA}_{3\,\text{h}}^{\text{IOP7}}$ and the $\text{CTRL}^{\text{IOP7}}$ analysis. Wind directions are also indicated by the green (resp. black) arrows for the $\text{CTRL}^{\text{IOP7}}$ (resp. $\text{RASTA}_{3\,\text{h}}^{\text{IOP7}}$) analysis. The data points assimilated in the $\text{RASTA}_{3\,\text{h}}^{\text{IOP7}}$ experiment until 07:30 UTC are also represented by the black data points. As expected, the analysis increments are well localised around the aircraft flight path. The assimilation of RASTA wind data has a large impact on the analysis since the increments can reach a value of approximately 12 m s$^{-1}$. The same behaviour is also seen when RASTA wind data are assimilated with smaller assimilation windows ($\Delta_t = 2$ h and $\Delta_t = 1$ h, not shown).

## 5.3   Verification against RASTA observations

Figure 6 (panels B to D) represents the wind speed differences of the $\text{RASTA}_{3\,\text{h}}^{\text{IOP7}}$ 1-, 2- and 3-h forecasts and the $\text{CTRL}^{\text{IOP7}}$ ones. At each forecast term, the black data points indicate the different RASTA locations which are available during a 1-h time window centred on the forecast time (forecast term $\pm$ 30 minutes). Figure 6 shows that, even though the increments are less organised as the forecast term increases, there is a noticeable impact of the assimilation of RASTA wind data on the subsequent forecasts at 07:00 UTC, 08:00 UTC and 09:00 UTC. Besides, some of the most substantial differences are co-located with RASTA locations (black data points in Figure 6) *and downstream of these locations.*

Figure 7 represents the standard deviation of the wind speed differences between RASTA observations and each experiment as a function of the forecast term. The standard deviations were calculated using all the data available within a 1-h time window centred on the forecast time (black data points in Figure 6). For instance, at 07:00 UTC, the 1-h forecast of each experiment are compared with the observations available between 06:30 UTC and 07:30 UTC. Similarly, at 08:00 UTC (09:00 UTC),

the 2-h (3-h) forecast of each experiment are compared with the observations available between 07:30 UTC and 08:30 UTC (08:30 UTC and 09:30 UTC).

As expected, the major differences between the different experiments appear on the analyses. The smallest standard deviation value is reached with the $\text{RASTA}_{1\,h}^{\text{IOP7}}$ experiment. Indeed, compared to the CTRL, the *standard deviation of the* wind speed has been reduced by a value close to 1.5 m s$^{-1}$. At the analysis time, the standard deviation values were calculated using the observations that were assimilated at the 06:00 UTC analysis in the $\text{RASTA}_{1\,h}^{\text{IOP7}}$ experiment (06:00 UTC + 30 minutes). As explained in the previous section, because of the non-rectilinear trajectory of the aircraft, the different RASTA analyses are slightly different. These differences explain why, when the comparison is performed against the observations which are available until 06:30 UTC , the standard deviation increases with increasing the length of the assimilation window. Nevertheless, in all three RASTA experiments, the standard deviation is always reduced in the analyses when RASTA observations are assimilated.

At 2- and 3-h term ranges, compared to the CTRL$^{\text{IOP7}}$, the assimilation of RASTA wind data leads to a systematic improvement in the standard deviation in the three RASTA experiments. By contrast, at 1-h term range, results indicate a negative impact of the assimilation of RASTA wind data since the three RASTA experiments exhibit larger standard deviation values. However, this negative impact should be taken cautiously since there are numerical noises and imbalances in the first two hours of integration due to spin-up in the AROME-WMed system (Seity et al., 2011).

Finally, Figure 7 demonstrates the benefit brought by the assimilation of RASTA wind data. Except at 1-h term range probably because of spin-up, there is an improvement in the predicted wind speed at all forecast term ranges. Nonetheless, it is hard to rank the different RASTA experiments. Similar results were also obtained in another case which occurred over sea on 11 October 2012 (not shown).

### 5.4 Impact on rainfall forecasts

The impact of the assimilation of RASTA wind data at 06:00 UTC is now illustrated on rainfall accumulation forecasts. To avoid the spin-up problem, the first hour of rainfall accumulation has been removed from the calculations. Figure 8 shows the 11-hour accumulated rainfall between 07:00 UTC and 18:00 UTC on 26 September 2012 (IOP7a) for the radar observations, CTRL$^{\text{IOP7}}$, the $\text{RASTA}_{1\,h}^{\text{IOP7}}$, the $\text{RASTA}_{2\,h}^{\text{IOP7}}$ and the $\text{RASTA}_{3\,h}^{\text{IOP7}}$ experiments.

In all experiments, the predicted rainfall accumulation patterns match well the observations. However, the maximum rainfall accumulation is much larger in the CTRL$^{\text{IOP7}}$ experiment (114 mm) than the observed one (76 mm). The $\text{RASTA}_{3\,h}^{\text{IOP7}}$ experiment is in much better agreement with the observations since the maximum rainfall accumulation has been reduced to a value close to 102 mm. *The assimilation of RASTA wind data with smaller assimilation windows (2 and 3 hours) does not significantly impact the rainfall forecasts. Indeed, the maximum rainfall accumulation are of same order of magnitude in*

*the RASTA$_{2h}^{IOP7}$ (113 mm) and RASTA$_{1h}^{IOP7}$ experiments (116 mm), compared to the CTRL$^{IOP7}$ one.*

The results in Figure 8 indicate a sensitivity to the choice of the assimilation windows. In particular, the best experiment is the one for which RASTA observations are assimilated with the larger assimilation window (RASTA$_{3h}^{IOP7}$). Therefore, in this specific case study, the rainfall accumulation forecasts are closer to the observations when more data are assimilated, even though some of them might no longer be valid at the assimilation time. This result can also be explained by the fact that horizontal wind components in moderately convective clouds are more representative of synoptic scales, and less likely to change as quickly as other meteorological variables, such as the humidity. However, this result may be only representative of this specific case study and should be taken cautiously.

## 6   Statistical study

The impact of RASTA wind data assimilation is now assessed over the 45-day cycled period during the HyMeX SOP1. Verification is first carried out against other assimilated observations types in subsection 6.1. Verification is then performed against rain gauges observations in subsection 6.2.

### 6.1   Comparison against conventional observations

Averaged over the 45-day experiment, the assimilation of RASTA wind data does not substantially impact the specific humidity and the temperature on both the analyses and the forecasts. Therefore, because the most significant differences only appear on the zonal (u) and meridional (v) wind components, results are only shown for the wind. Calculations are not shown for the analyses but only for the 3-h forecasts. Indeed, since the observations used for the comparisons are all assimilated, the fit to observations is better in CTRL$^{SOP1}$ than in the RASTA experimental runs.

Because RASTA wind data is limited in space around the Mediterranean area (see black lines in Figure 1) and depends on the presence of cloud or precipitation along the aircraft flight path, its assimilation impact is also limited in space. Hence, at each assimilation time, a RASTA-limited validation area is employed. It contains the aircraft flight path $\pm 0.5°$ both in latitude and longitude. Only the conventional observations (commercial aircraft data, radiosonde and profiler) which are available in the RASTA-limited area are used for the calculations. Since the assimilation impact of RASTA wind data is also limited in time, calculations are only performed over the 35 runs in which RASTA wind data were assimilated with the largest assimilation window. Figure 9 shows the differences in standard deviation error for 3-h wind forecasts between the CTRL$^{SOP1}$ experiment and the RASTA$_{3h}^{SOP1}$ (red), the RASTA$_{2h}^{SOP1}$ (blue) and the RASTA$_{1h}^{SOP1}$ (green) experiments. Negative (positive) differences indicate a positive (negative) impact of the assimilation of RASTA wind data. The total number of observations used for the calculation is represented by the black "+"s in the top x-axis.

In general, Figure 9 indicates that the impact of the assimilation of RASTA wind data on the 3-h forecasts is hard to assess. Indeed, compared to commercial aircraft wind observations (left panel), the vertical profiles of the standard deviation demonstrate a neutral impact. However, compared to radiosonde (middle panel) and profiler (right panel) observations, there is a slight positive to negative impact depending on the assimilation window, which is probably a deluding effect due to the small number of conventional observations available in the area of interest. The comparison with ground-based radar data gives similar results (not shown).

## 6.2 Impact on rainfall forecasts

Forecast scores against rainfall measurements are now calculated over the 35 runs (out of 360) in which RASTA data were assimilated with the largest assimilation window. The verification is conducted using the rain gauge network available from the HyMeX database (doi:10.6096/MISTRALS-HyMeX.904) , whose locations are indicated by the blue markers in Figure 1. For the comparisons, model outputs are interpolated to the rain gauges station locations using a linear interpolation. Model outputs and rain gauge measurements are then averaged in boxes of $0.25° \times 0.25°$ within each RASTA-limited validation area.

Categorical scores have been calculated: Heidke Skill Score (HSS), Probability Of Detection (POD) and False Alarm Ratio (FAR). To avoid the spin-up problem, the first hour of rainfall accumulation has been removed from the calculations. The HSS, POD and FAR of the 8-h accumulated rainfall forecasts for the three RASTA experiments are displayed in red in Figure 10 as a function of the rainfall accumulation threshold (mm). The scores of the $CTRL^{SOP1}$ experiment are also shown in black. The bootstrap confidence intervals are displayed for each threshold by the dashed lines. The impact of the assimilation of RASTA wind data is positive if the red lines are above (below) the black ones for the HSS and POD (FAR).

Figure 10 shows that the general pattern is similar for the three RASTA experiments, which indicates that the choice of the assimilation window does not impact significantly the subsequent forecasts. Even though the bootstrap confidence intervals increase with the threshold, differences with the $CTRL^{SOP1}$ experiment are more pronounced at larger thresholds in any of the three RASTA experiments. The most significant differences appear for the $RASTA_{3h}^{SOP1}$ and $RASTA_{2h}^{SOP1}$ experiments, which is consistent the results found for the IOP7a case study in subsection 5.4. In addition, except for the $RASTA_{1h}^{SOP1}$ experiment, the assimilation of RASTA wind data tends to improve slightly the scores above approximately 10 mm.

It should be noted that this slight positive improvement of the heavier rainfall can also be seen for the 8- and 11-h forecasts (not shown). Finally, the benefit brought by the assimilation of RASTA wind data decreases with the forecast term range ($\geq$ 11-h forecasts), which is partly explained by the lateral boundary conditions. Indeed, after a few hours, the increments are replaced by inputs from the same coupling model.

## 7  Discussions and conclusions

This article reports on the first study in which vertical profiles of wind measured by vertically-pointing airborne Doppler W-band radar are assimilated in a kilometre-scale NWP model. The study was performed in a quasi-operational framework with a special version of the Météo-France operational kilometre-scale model AROME with its 3DVar assimilation system. The data were provided by the airborne Doppler W-band radar RASTA during a 45-day period over a region of the Mediterranean area very prone to heavy rainfall. RASTA is a multi-beam antenna system that can retrieve the three components of the wind fields, which allows the direct assimilation of the retrieved horizontal wind components.

A sensitivity study to the choice of the assimilation window was performed. Indeed, RASTA wind data are not measured simultaneously at the assimilation time, but over the flight leg. Consequently, at the assimilation time $T$, the 3DVar assimilation system of AROME-WMed ingests all data available along the aircraft path during the assimilation window $\Delta_t$, as if they were valid at time $T$. Therefore, the ability of RASTA wind data to improve short-term forecasts of heavy precipitation events was tested with three different assimilation windows $\Delta_t$: three hours (RASTA$_{3\,h}$), two hours (RASTA$_{2\,h}$) and one hour (RASTA$_{1\,h}$).

The positive impact of the assimilation of RASTA wind data was first evidenced in a case of heavy rainfall, which occurred during the Intensive Observation Period 7a (IOP7a) on 26 September 2012. This case study was selected because the data that are assimilated at the 06:00 UTC analysis are located upwind from where the heavy rainfall took place. Such a configuration is required to study a potential impact of the assimilation of RASTA wind data on the subsequent forecasts. Except at very short-term range (one hour) because of spin-up, the assimilation of RASTA wind data led to a systematic improvement of the predicted wind at all short term ranges (two and three hours) in any of the three RASTA experiments. It could be interesting to repeat the same study with the more recent operational AROME system because the model spin-up has been reduced to less than 1 hour (Brousseau et al., 2016). Besides, the 11-h accumulated rainfall forecasts are also in much better agreement with the observations. Therefore, this case study demonstrates a positive impact of the assimilation of RASTA wind data to better predict this rainfall event. Similar results were also obtained for another case which occurred over sea on 11 November 2012 (not shown in this article).

A cycling data assimilation experiment has also been conducted over a 45-day period from 24 October 2012 to 05 November 2012, for the CTRL experiment and for the three RASTA data assimilation experiments. The comparisons against other assimilated observations and rain gauges measurements indicate an overall neutral impact, which is probably due to the small percentage of RASTA wind data which were assimilated among the total number of observations. Nevertheless, results of this statistical study are encouraging since no major detrimental effect was found and a slight positive improvement in the 5-, 8- and 11-h precipitation forecasts of heavier rainfall was evidenced.

The sensitivity study to the assimilation window on the IOP7 case study and on the statistical study suggested that the quality of the rainfall accumulation forecasts increases with the length of the assimilation window. Hence, it seems preferable to assimilate more data to have larger coverage by increasing the length of the assimilation window. However, results should be taken cautiously since the sensitivity study was only conducted over 35 analysis cases. More cases should be explored over other field campaigns to corroborate the results of this sensitivity study. Besides, the issue of the length of the assimilation window becomes less important if the assimilation frequency increases and/or a shorter assimilation cycle is used, such as in the new AROME system (Brousseau et al., 2016).

It is probable that low quality data did pass the quality control, and were thus assimilated. Zhang et al. (2012) show the importance of specifying a strong data quality control. Hence, a more efficient data quality control should improve our results. Finally, another perspective is to assimilate the W-band radar reflectivity jointly with RASTA wind data to study if modifying the thermodynamic and the dynamic state of the model in a consistent way in the initial state would lead to more significant improvements. Indeed, Janisková (2015) demonstrated a slight positive impact of the assimilation of W-band space-borne radar using a 1D+4D-Var technique. The 1D+3DVar assimilation method that is operationally used to assimilate the radar reflectivity in AROME (Caumont et al., 2010; Wattrelot et al., 2014) will be employed to assimilate the W-band reflectivity.

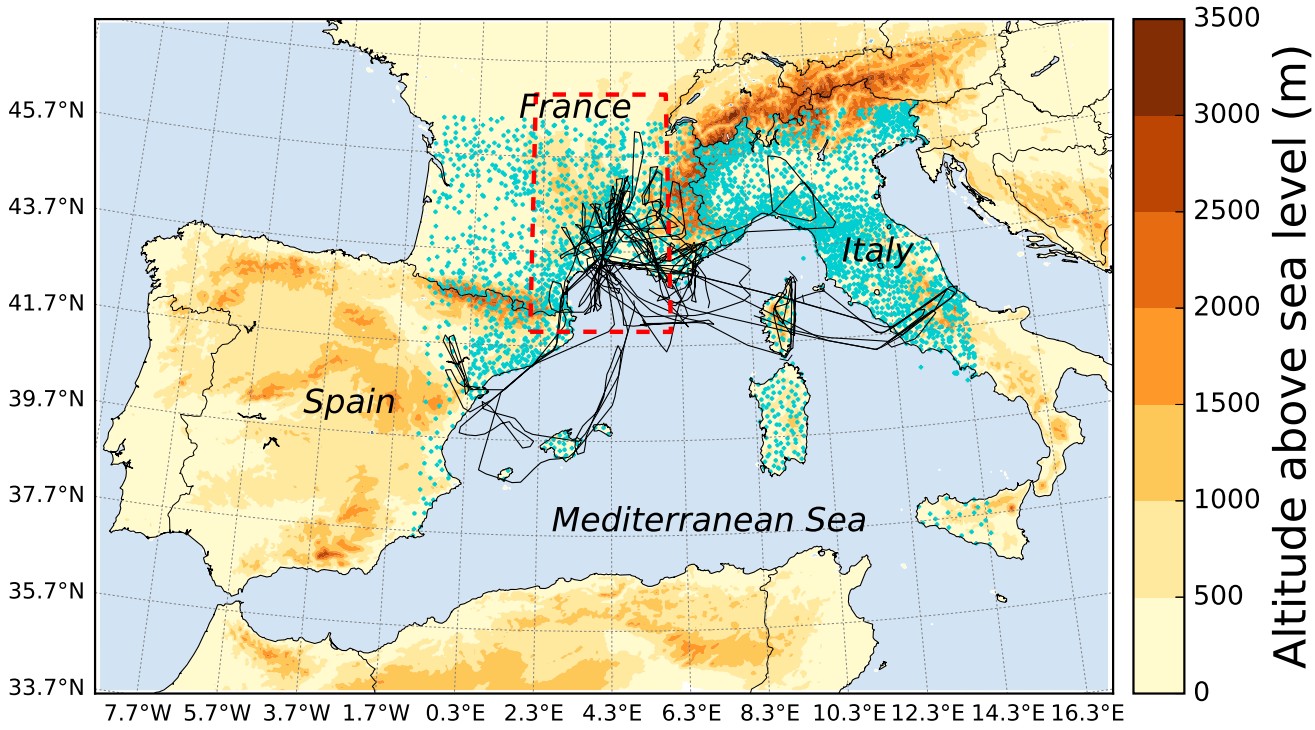

**Figure 1.** The Falcon 20 flight paths (black lines) during the HyMeX first Special Observing Period over the AROME-WMed domain. The altitude of ground above sea level (in metres) is represented by the colour shades. Rain gauges are represented by the blue markers. The area surrounding the IOP7a case study is indicated by the red box.

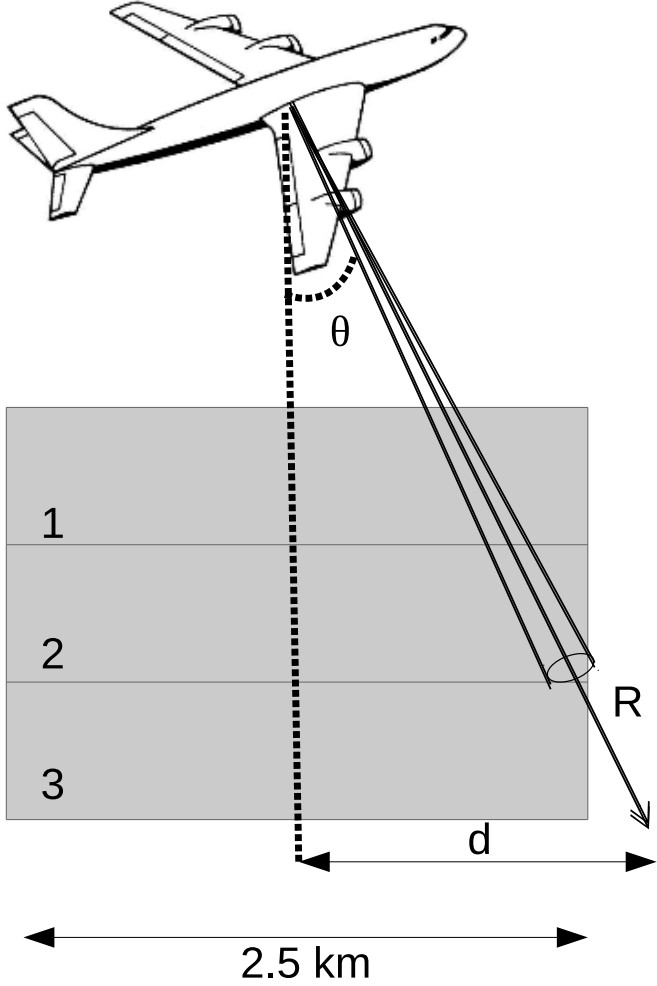

**Figure 2.** Schematic view of the aircraft to represent the data which are taken into account to calculate the "super-observations". If the d is larger than $\frac{2.5}{2}$ km, the data is not used to calculate RASTA "super-observation". In this configuration, the observation is not used to in cell number 3.

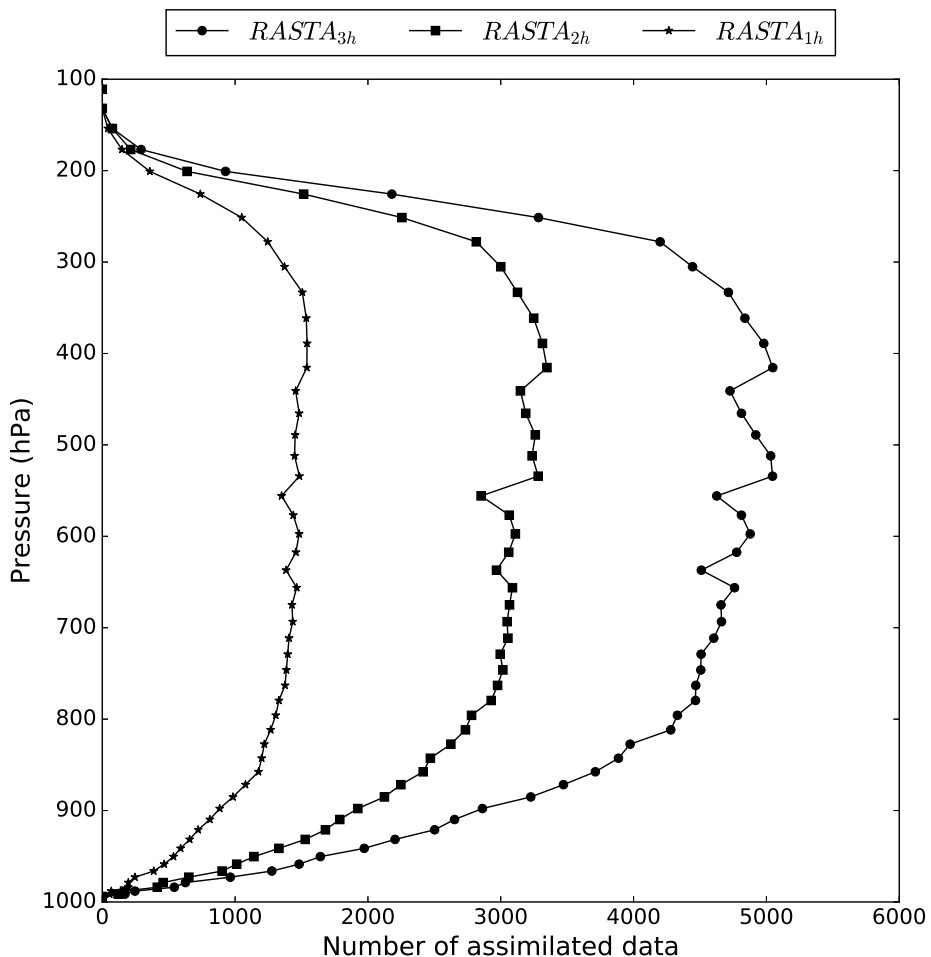

**Figure 3.** Number of RASTA horizontal wind data that are assimilated as a function of the pressure for the RASTA$_{3\,\text{h}}^{\text{SOP1}}$, RASTA$_{2\,\text{h}}^{\text{SOP1}}$, $RASTA_{1h}$.

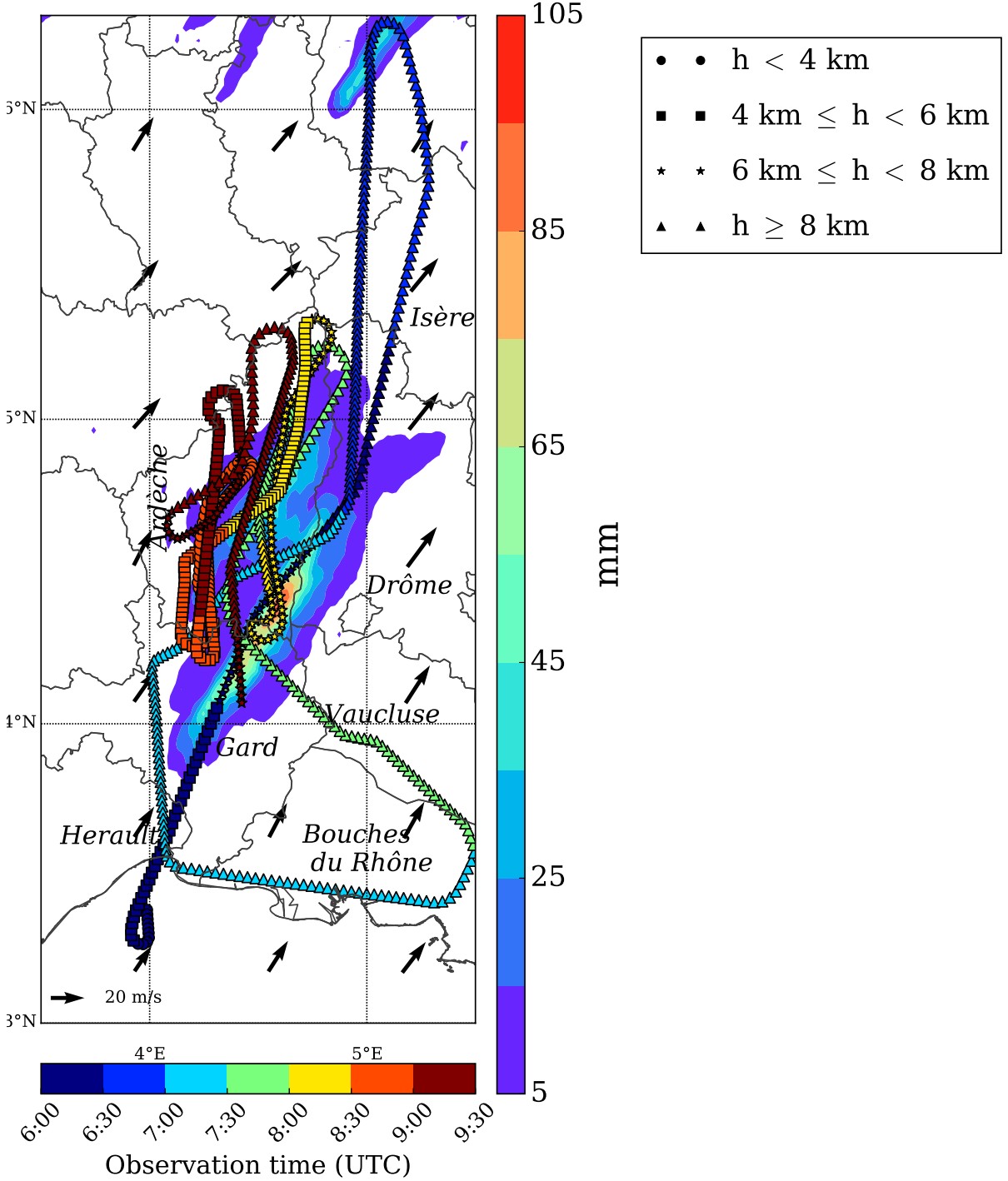

**Figure 4.** Area under the red box in Figure 1. 24-hour accumulated rainfall observed by weather radar between 00:00 UTC on 26 September and 00:00 UTC on 27 September are represented by the shadings (scale on the right). The observation time along the Falcon 20 flight path is indicated by the colour data points (scale on the bottom). The circle, square, star and triangle markers indicate aircraft's altitude (see legend). Arrows represent the wind direction from the CTRL$^{IOP7}$ analysis at 06:00 UTC at approximately 4 km of altitude (model level 30).

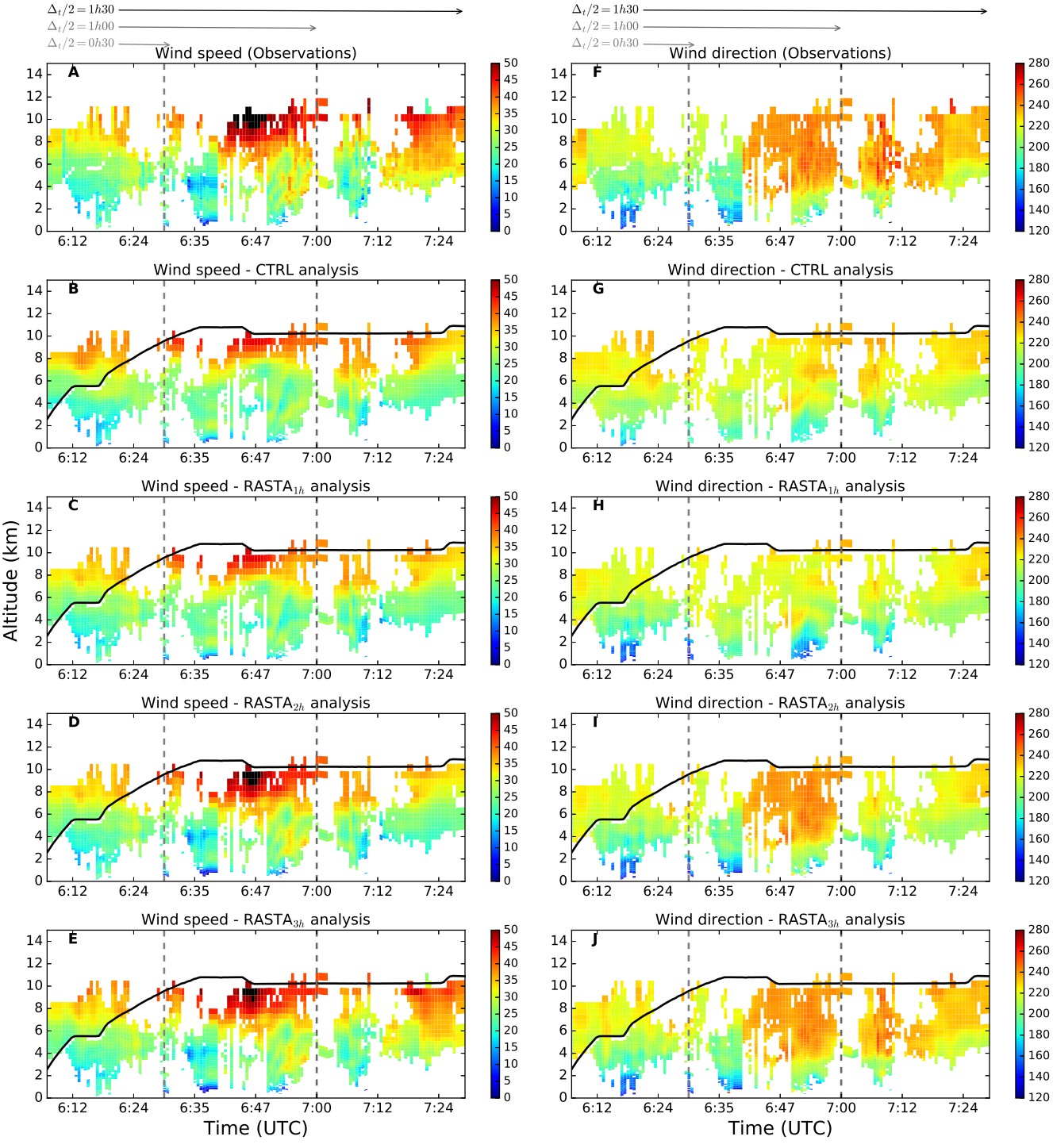

**Figure 5.** Wind speed (A to E panels) and wind direction (F to J panels) for (from the top to the bottom) the observations, the CTRL[IOP7], the RASTA$_{1h}^{IOP7}$, the RASTA$_{2h}^{IOP7}$ and the RASTA$_{3h}^{IOP7}$ 06:00 UTC analyses on 26 September 2012 (IOP7a). The three different assimilation windows $\Delta_t$ are delimited by the vertical lines. Aircraft's altitude above sea level is represented by the black line.

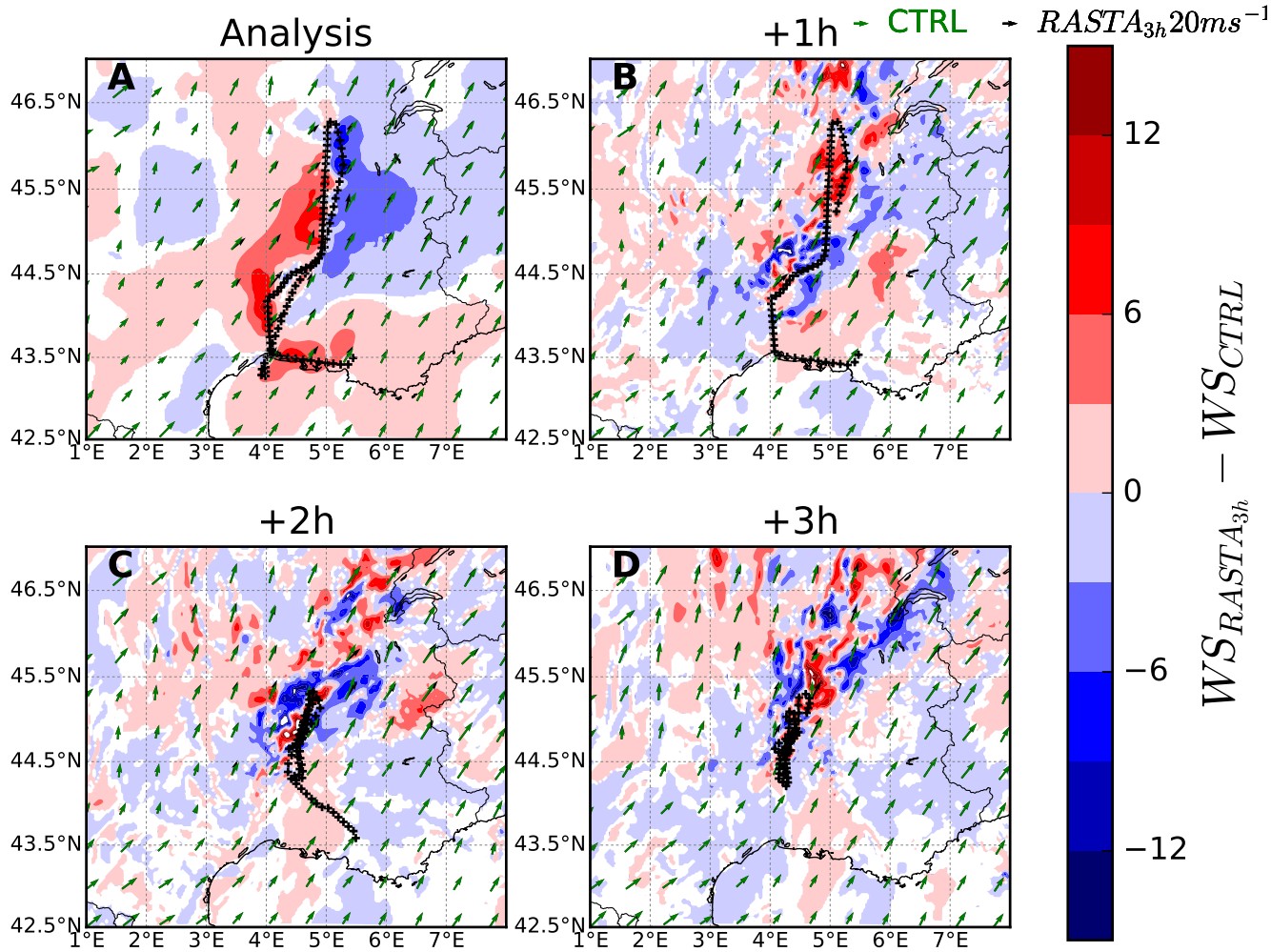

**Figure 6.** Wind increments between the RASTA$_{3h}^{IOP7}$ and the CTRL$^{IOP7}$ experiments for the analysis at 06:00 UTC and for the 1-, 2-, and 3-h forecasts at 07:00 UTC, 08:00 UTC and 09:00 UTC 26 September 2012 (IOP7a) at approximately 4 km of altitude (model level 30). The black data points represent the location of RASTA data within a 1-h time window centred on the forecast term range.

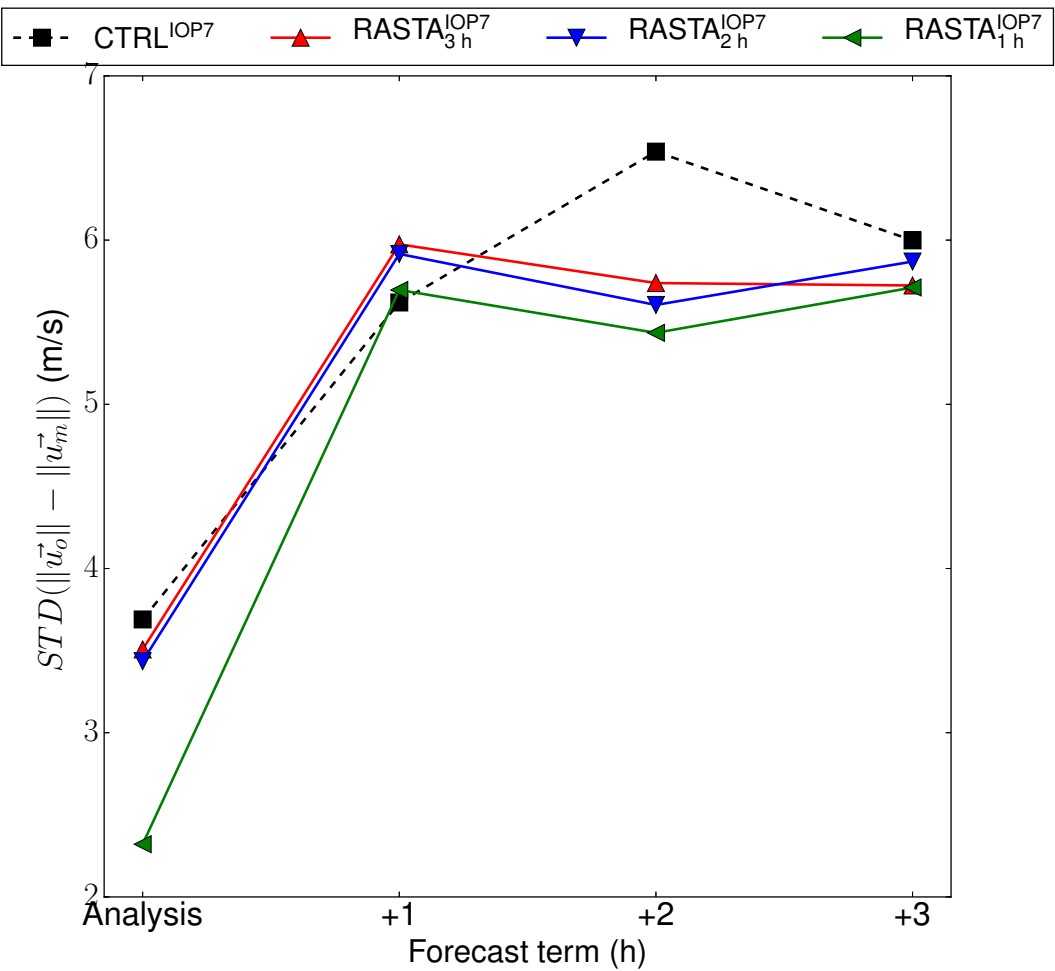

**Figure 7.** Standard deviation of the wind differences between RASTA observations and each experiment (see legend) as a function of the forecast term from the 0600 UTC analysis 26 September 2012 (IOP7a).

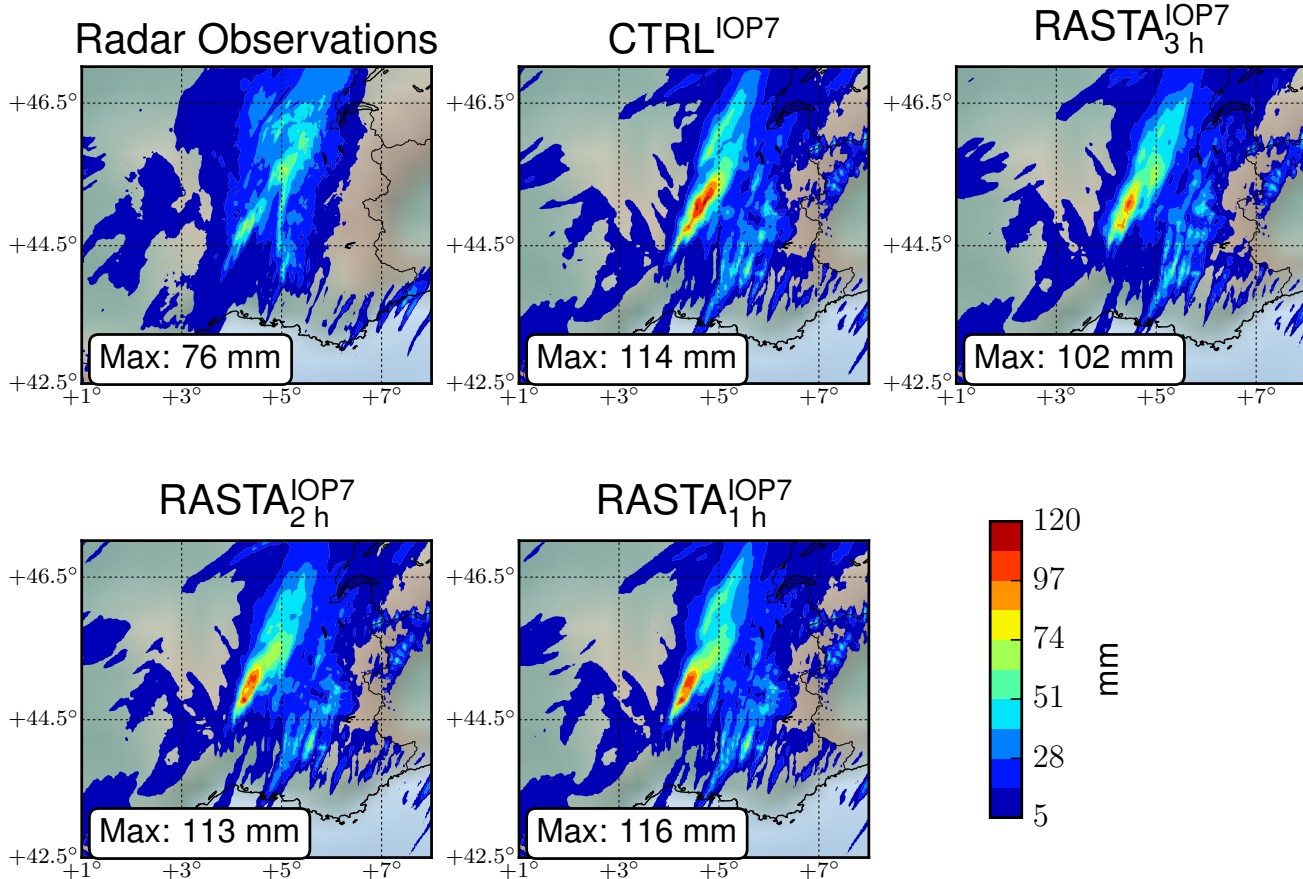

**Figure 8.** 11-h accumulated rainfall between 07:00 UTC and 18:00 UTC 26 September 2012 (IOP7a) for radar observations, the CTRL[IOP7], RASTA$_{3\,h}^{IOP7}$, RASTA$_{2\,h}^{IOP7}$, RASTA$_{1\,h}^{IOP7}$ experiments.

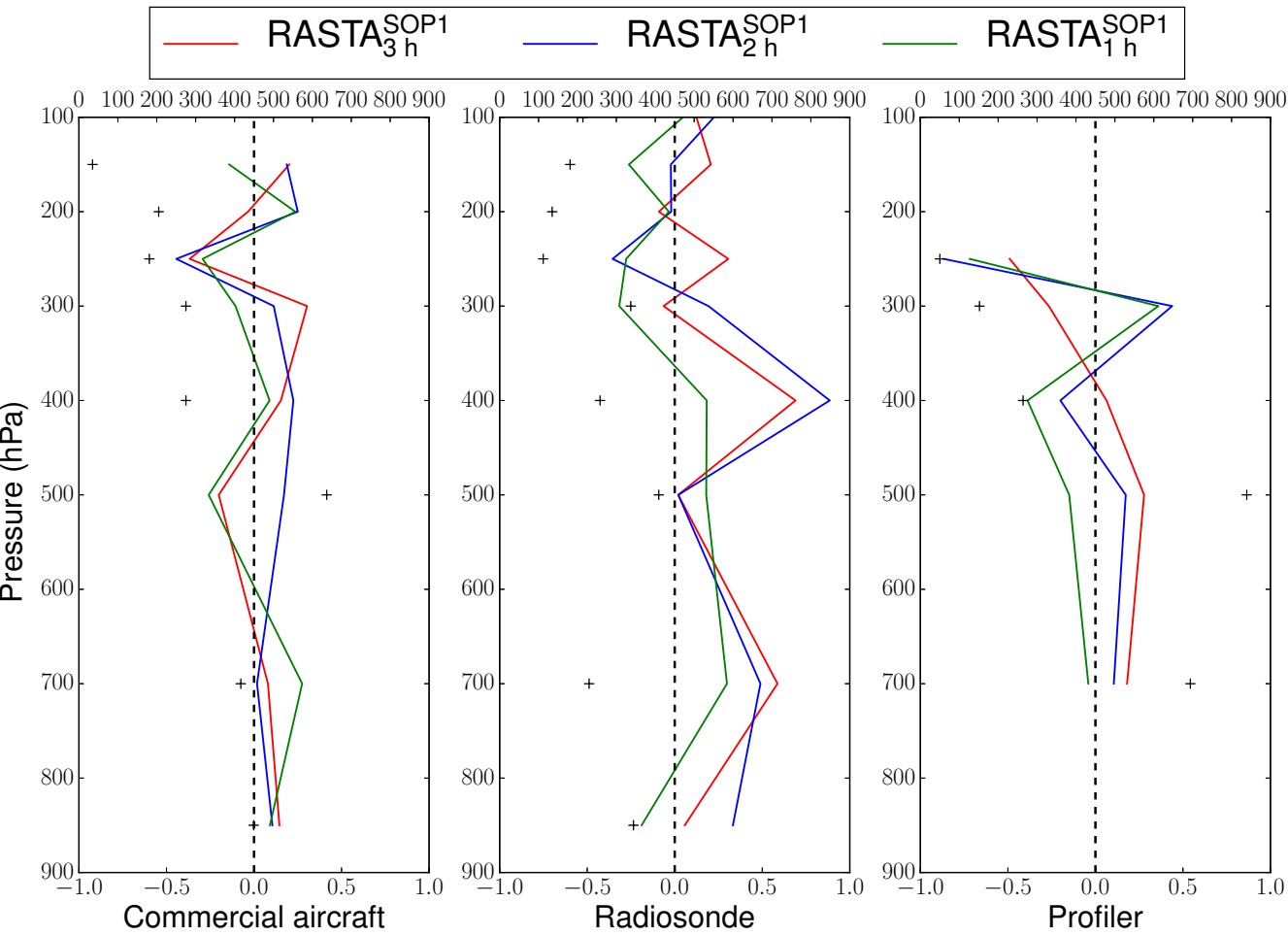

**Figure 9.** Differences of standard deviation error for 3-h wind forecasts between the CTRL$^{\text{COP1}}$ experiment and the RASTA$^{\text{SOP1}}_{3\,\text{h}}$ (red), the RASTA$^{\text{SOP1}}_{2\,\text{h}}$ (blue) and the RASTA$^{\text{SOP1}}_{1\,\text{h}}$ (green) experiments. Negative differences indicate a positive impact of the assimilation of RASTA wind data. The standard deviation errors are computed for commercial aircraft (left panel), radiosonde (middle panel) and profiler (right) observations. All the scores are computed over the 35 runs in which RASTA wind data were assimilated with the largest assimilation window over the RASTA-limited area. In each panel, the number of observations used for the calculation is represented by the black data "+"s in the top x-axis.

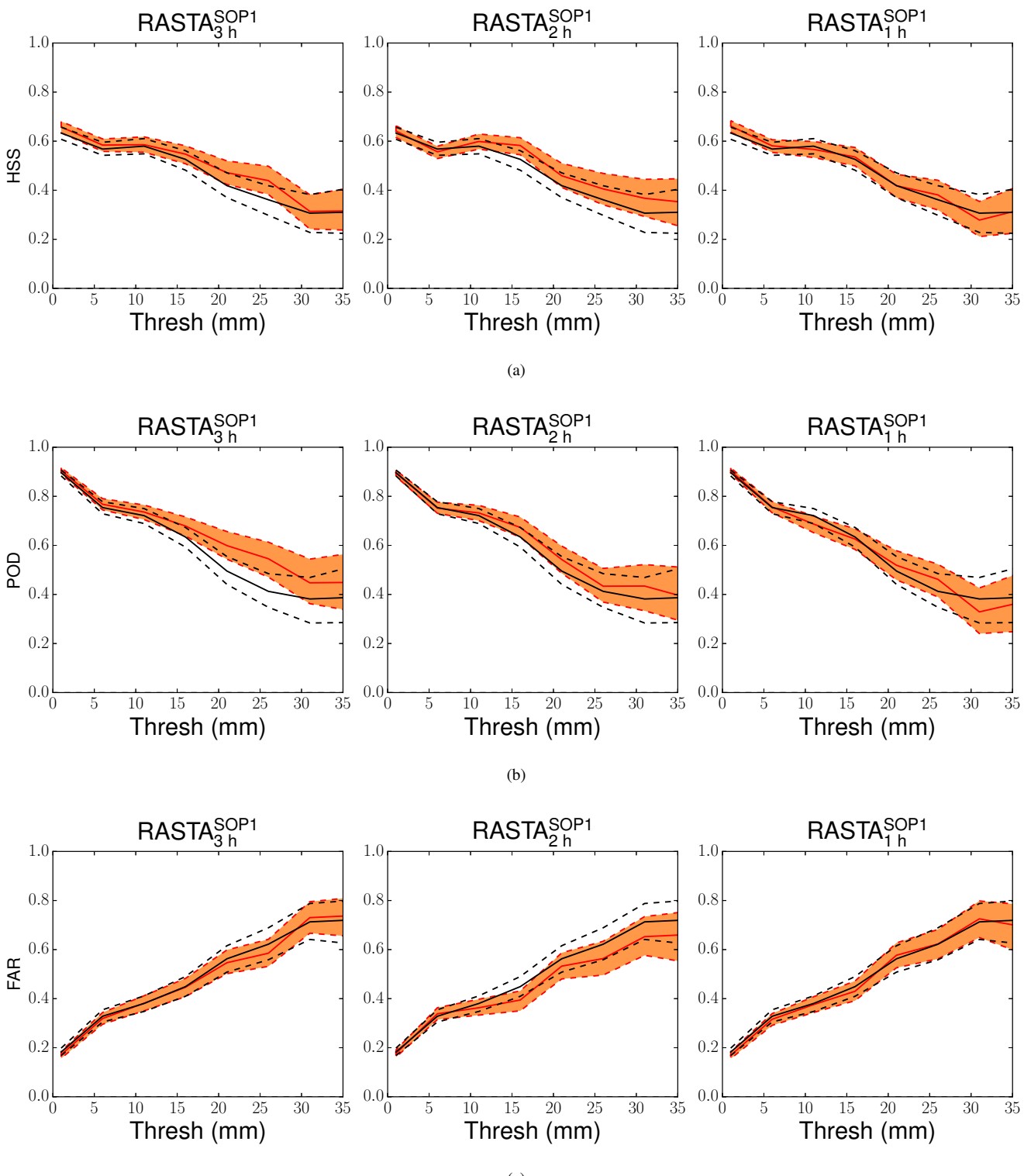

**Figure 10.** HSS (left panels), POD (middle panels) and FAR (right panels) of the 9-h cumulated precipitation forecasts versus rain gauge measurements for the three RASTA experiments (in red) and for the CTRL$^{SOP1}$ experiment (in black). Calculations were performed over the 35 runs in which RASTA wind data were assimilated with the largest assimilation window. The error bars (dashed lines) represent the 90% bias-corrected and accelerated (BCa) bootstrap confidence intervals (see Efron et al., 1993).

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

*Competing interests.* The authors declare that they have no conflict of interest.

*Acknowledgements.* This work is a contribution to the HyMeX program supported by MISTRALS, ANR IODA-MED Grant ANR-11-BS56-0005 and ANR MUSIC Grant ANR-14-CE01-0014. This work was supported by the French national programme LEFE/INSU. The authors acknowledge the DGA (Direction Générale de l'Armement), a part of the French Ministry of Defense, for its contribution to Mary Borderies's PhD. The authors thank SAFIRE for operating the French Falcon 20 research aircraft during HyMeX-SOP1. The authors are grateful to Pierre Brousseau for his technical help. Two anonymous reviewers are also gratefully acknowledged for their comments which helped to significantly improve the quality of the article.