# Peer review of "Assimilation of wind data from airborne Doppler cloud-profiling radar in a kilometre-scale NWP system"

_Natural Hazards and Earth System Sciences, 2018_

## Referee Comment (RC1) · Anonymous Referee #1 · 4 Oct 2018

Review comments on "Assimilation of wind data from airborne Doppler cloud-profiling radar in a kilometre-scale NWP system" by Borderies et al.

The authors assimilated wind profile data observed by an airborne Doppler radar, and then examined their impacts on wind field and rainfall forecasts using a kilometer-scale DA. Positive impacts were seen both in the wind field and rainfall forecasts of the case study. Especially they found slightly better result in the 3-h assimilation window experiment. On the other hand, a statistical study showed mostly neutral skills between the 1-h, 2-h, and 3-h assimilation windows, even in comparison with the control experiment (no data assimilation). The 1-h assimilation window experiment was very slightly hope-
ful in rainfall forecast. I recommend the authors to improve the manuscript and conduct some additional investigations before publication.

[Major comments]

1. The authors conducted two verification of a case study and a statistical examination. The case study showed that the RASTA assimilation improved wind and rainfall fields, and the 3-h assimilation looked best. On the other hand, the statistical examination for the entire domain in Figure 1 illustrated that the RASTA assimilation mostly had no impact even on the wind field, and only the 1-h assimilation has some skill in rainfall forecasts. Since the RASTA data is limited in cloud region, the assimilation impact is also limited in time and space. I suggest that the statistical examination is re-conducted over a limited area, for instance, the Figure 2 area, or convective-system-related area, or RASTA-related area, since the inconsistent results between the case study and the statistical examination makes the readers confused.

2. I agree that Figure 6 implies a spin-up problem in forecast. For the reason of the spin-up, I doubt that the observational error of RASTA use in the present study would be smaller than appropriate value because it is the same with that of radiosondes. The error should be larger since RASTA includes much more sources of errors than radiosondes include.

3. I suggest that Figure 5, and explanations for Figures 4 and 5 will be modified. The increment in Fig. 5A is reflected the flight path of all observations, thus, all data points assimilated in this 3-h window should be presented in Fig. 5A. Moreover, the 1-h, 2-h, and 3-h assimilation window experiments include the observation until 0630, 0700, and 0730 UTC, respectively (L10 P7). I think that this different time limitations create the difference between panels in Figure 4 unlike the authors explanation on overpasses (L6-17 P8). Please exam and discuss this point of view.

4. It is amazing for me that RASTA\_3h in Figure 4 improved the wind filed even at the end of the assimilation window because the experiment did not employ FGAT. Since
the RASTA data only exist in cloud area, 3 hours seems too long to assimilate the data appropriately. I understand that this is the motivation of the authors to conduct three experiments. If they use FGAT, the 3-h experiment may significantly improve the result. I recommend the authors to conduct the FGAT experiment additionally if possible.

5. The authors used the median value of observations in a grid box (L16 P5) for thinning. If observational data distribute followed the Gaussian PDF and their number are large enough, the median and the mean values are the same. Usually "super observations" are made by the "mean" method in order to reduce representativeness errors and avoid noises. Therefore, the authors should explain why they adopt the "median" method instead of the mean.

6. English needs to be proofread by professional native speaker(s) with scientific background.

[Minor comments]

L23 P1: "To fill the gap in clear air condition" I suggest the authors to refer the following articles, because wind observations in clear air can be also provided by Doppler lidars (air-born and ground-based), and clear air echoes (insects) by Doppler radars.

[Ground-based lidar] Kawabata, T., H. Iwai, H. Seko, Y. Shoji, K. Saito, S. Ishii, and K. Mizutani, 2014: Cloud-Resolving 4D-Var Assimilation of Doppler Wind Lidar Data on a Meso-Gamma-Scale Convective System. Mon. Wea. Rev., 142, 4484–4498, doi: 10.1175/MWR-D-13-00362.1. [Air-born lidar] Weissmann, M., R. H. Langland, C. Cardinali, P. M. Pauley, and S. Rahm, 2012: Influence of airborne Doppler wind lidar profiles near Typhoon Sinlaku on ECMWF and NOGAPS forecasts. Quart. J. Roy. Meteor. Soc., 138, 118–130, doi:10.1002/qj.896. [Clear air echoes] Kawabata, T., H. Seko, K. Saito, T. Kuroda, K. Tamiya, T. Tsuyuki, Y. Honda, and Y. Wakazuki, 2007: An assimilation and forecasting experiment of the Nerima heavy rainfall with a cloud-resolving nonhydrostatic 4-dimensional variational data assimilation system. J. Meteor. Soc. Japan, 85, 255–276, doi:10.2151/jmsj.85.255.

NHESSD
L19 P2: "has never been investigated" I did not understand what thing has never been investigated in the following "vertical profiles from Doppler W-band radar". "vertical profiles from Doppler radar"? "W-band radar"? "vertical profiles" by W-band radar? ("horizontal" winds have been done)? Please clarify.

L30 P2: "first" This is the same with the above. What is the first?

L8 P3: "HyMeX-SOP1" What is this? Spell out it and add explanation.

L28 P3: "six Cassegrain antennas" How do these six antennas observe three directions above and below the aircraft? Add explanation and, if possible, a schematic figure.

L30 P3: "unambiguous distance" "unambiguous velocity" What are these? Observational range and available observations? But, in Figure 4, we see larger observations than 7.8 m/s.

L22 P4: "2.5 km x 2.5 km" Modify it to 2.5 x 2.5 km" and add the number of horizontal grids or the horizontal size of the domain.

L25 P4: "specially designed" What is the special in this study? Please clarify.

L7 P5: "GPS" Spell out it. GPS stands for Global Positioning System operated by U.S.A.. I guess the authors use other navigation satellite systems like Galileo and GLONASS. In this case, GPS should be replaced by "GNSS" (Global Navigation Satellite System).

L17-19 P5: "When the aircraft – removed from the interpolation." It is hard to understand the situation and removed data. Did the authors remove the data only outside the grid box or the whole profile of the data? It should be better to show a schematic figure of the aircraft with the six radar antennas, and wind profiles in and out the grid boxes.

L29 P6 and L10 P10 I suggest that the title of Section 5 and 6 as well as the examinations are named as "the case study" and "the statistical study" instead of IOP7a and
HyMeX SOP1, respectively.

L30, L31, L34 P9: "the maximum rainfall" Please show the exact maximum values in each experiment, not approximated values.

L31-31 P10: "small number" From Figure 8, the numbers of observations are several thousands. These are not "small".

Figure 1 Add the explanation on the red box.

Figure 3 It is helpful for the readers if the authors add the information on flight level in this figure, for instance, by changing the size of circles as height, or by replacing the circles with triangles or rectangular or cross-marks as height.

Figure 4 Add (a), (b), (c) and etc. or figure titles to each panel to refer it easier.

Figure 7 Add the maximum rainfall amount values to each panel.

Figure 8 I suggest that Figure 8 will be illustrated by the difference between CTRL and others, not each profile, in addition to the examination on the limited area (see the major comment).

---

## Referee Comment (RC2) · Anonymous Referee #2 · 22 Oct 2018

Review comments to the paper "Assimilation of wind data from airborne Doppler cloud-profiling radar in a kilometre-scale NWP system", manuscript nhess-2018-246.

This paper describes the assimilation of airborne W-band radar observations in a km-scale NWP model. The observations are collected from aircraft flights but the main purpose of the paper however, is to introduce this observation type in preparation for satellite based observations. This is why the paper is important and interesting. However, to draw any strong conclusions from this study, e.g. whether a short or long assimilation window is more beneficial is too wild I think. The results are also rather inconclusive as shown by the authors. The reason is that the observations are very

limited both in time and space. The functionality and positive results however are very promising. My opinion is that the paper is well written and worth publishing when the comments below are taken into consideration.

Comments: Page 2, line 16: "...at each kilometer levels..." Please explain better what this means?

Page 2, line 35: What is different compared to the operational version? Just a brief, short explanation would be good.

Page 3, line 30: 7.8 m/s is a rather low value for the unambiguous velocity. In fact this is one of the main challenges in Doppler wind assimilation. A de-aliasing, or unfolding algorithm can work fine for unfolding once but what if the wind speed is high enough to fold twice? Then it will be much more uncertain. Are the authors confident with the algorithm used and/or that there are no wind speed above this limit? Another complicating factor is that the aircraft is moving and this also needs to be taken into account in the unfolding. Perhaps it is outside the scope of the paper to discuss this in detail but a brief discussion about this is necessary since it is crucial when using the data.

Page 5, line 21: "every 3 time steps" What does this mean?

Page 5, lines 25-28: It could also be so that "important" data is collected late (or early) in a longer assimilation window if the aircraft e.g. flies into a convective cell. In studies like this it would also be very beneficial to run with FGAT (First Guess at Appropriate Time) or even better using a 4D-Var assimilation scheme.

Page 6, line 10: The same observation error as radiosondes. Isn't this a bit optimistic?

Page 6 lines 11-13: Is there any other quality control applied to the observations? If yes, what and how. If no, why not?

Page 6, lines 20-24: The last two sentences in the paragraph is really hard to follow. Please re-write to make it more clear.

Page 7, table 1: In the column of assimilated data it says Conventional and Conventional + RASTA. Are only conventional observations assimilated apart from the RASTA observations? In section 3.2 there are many more observations mentioned that are not consider to be conventional, e.g GNSS and satellite data.

Page 7, lines 10-14: The data collection starts at 06:10 and the analysis time chosen to study is 06:00. This means that the 3 hour window only will be a 1.5 hour window. Is there no better example where one can find a data collection more centered around the analysis time. Why not show an example from 09:00? Then the data collection will also be skewed but there will at least be data available on both sides of the analysis time.

Page 7, line 20: "...expected to improve the forecast..." Is this really the case? It depends on how the data is introduced, observation errors and how the model performed without the data.

Section 5.2: It would be interesting to see the same case in a cycled period. If the cycled run builds up its own "climate" could the results be even better?

Page 8, lines 19-25: Please explain figure 5 better.

Page 9, line 9: The observations assimilated are not from $\pm30$ minutes from 06 UTC. They are from +30 minutes. Right?

Page 9 and figure 6: This is a typical behavior when observations are assimilated with a too small observation error. The analysis is adjusted to fit the RASTA observations too much but as soon the model starts running it adjust itself to its own more comfortable state. The analysis will look very good, especially compared to RASTA observations, but there will be a spinup to the model state as seen in the figure. Why not run the same experiment with different observation errors too see if that can reduce the spinup and improve the forecasts, not only the analysis?

Page 9, line 28: What forecast lengths are used for the 12 hour accumulation? Inter-

esting to know in view of the above discussion about spinup.

Page 10, line 18: Calculations are only performed over the 35 runs (I assume that this means analysis times) with RASTA observations. Why?

Page 13, line 11: Again quality control is mentioned and that it is important. What quality control was applied to the data assimilated here (see also comment above)?

---

## Author Comment (AC1) · 24 Jan 2019

We thank Reviewer 1 for his/her constructive comments. Please, you can find attached our responses, along with the revised version of the article (changes have been made in red in the text).

Please also note the supplement to this comment:
https://www.nat-hazards-earth-syst-sci-discuss.net/nhess-2018-246/nhess-2018-246-AC1-supplement.zip

2018-246, 2018.

---

## Author Comment (AC2) · 24 Jan 2019

We thank Reviewer 2 for his/her constructive comments. Our responses are in the attached file, along with the revised version of the article (changes are in red).

Please also note the supplement to this comment:
https://www.nat-hazards-earth-syst-sci-discuss.net/nhess-2018-246/nhess-2018-246-AC2-supplement.zip

---

## Author Response (AR1)

Reviewer:1

We thank Reviewer 1 for his/her constructive comments. Our responses are given below in red.

Responses to the major comments:

1. The authors conducted two verification of a case study and a statistical examination.

The case study showed that the RASTA assimilation improved wind and rainfall fields, and the 3-h assimilation looked best. On the other hand, the statistical examination for the entire domain in Figure 1 illustrated that the RASTA assimilation mostly had no impact even on the wind field, and only the 1-h assimilation has some skill in rainfall forecasts. Since the RASTA data is limited in cloud region, the assimilation impact is also limited in time and space. I suggest that the statistical examination is re-conducted over a limited area, for instance, the Figure 2 area, or convective-system-related area, or RASTA-related area, since the inconsistent results between the case study and the statistical examination makes the readers confused.

The authors are grateful to Reviewer 1 for his comment because now we have a better consistency between the case study and the statistical examination. The statistical examination has been re conducted over a RASTA-limited area. This area contains the aircraft flight path +/- 0.5° both in longitude and latitude. The RASTA-limited validation domain is larger than the exact flight path because the increments are advected as the forecast term increases. The text has been modified in section 6.1.

The comparison against conventional observations indicates similar results (see section 6.1 and Figure 9 of the revised version): generally the impact is slightly negative to slightly positive. Besides, the differences are less than 0.5 m/s, so the impact is neutral.

The methodology employed to compute the scores against rain gauge measurements has been modified. In the RASTA-limited validation area, observations and model outputs are first averaged in boxes of 0.25°\*0.25°, and then concatenated over the 35 assimilation cases. Bootstrap confidence intervals are calculated with these new sets of observations/model outputs. To avoid the spin-up problem, the first hour of rainfall accumulation has also been removed from the calculations.

- The new results are more consistent with the case study: the best scores are reached with the largest assimilation windows (2h or 3h) and the most significant differences appear with the RASTA\_3h and RASTA\_2h experiments.
- Generally, the impact is slightly positive to neutral. The use of the smallest assimilation window leads to the most neutral impact, which is also consistent with the IOP7a case study.
- In the previous version of the paper, the differences between the CTRL experiment and the RASTA experiments appeared above approximately 25 mm. Now in Figure 9 (Figure 10 in the revised version) we can see differences above 10 mm.
- Figure 9 has been modified (Figure 10 in the revised version), together with the text in sections 6.2, 7 and in the abstract.

2. I agree that Figure 6 implies a spin-up problem in forecast. For the reason of the spin-up, I doubt that the observational error of RASTA use in the present study would be smaller than appropriate value because it is the same with that of radiosondes. The error should be larger since RASTA includes much more sources of errors than radiosondes include.

We are not sure that the observation error should be larger than the one used for radiosondes.

Indeed, RASTA wind data during the HyMeX-SOP1 field campaign have been compared against ground-based Doppler radar by Bousquet et al. (2016). Results of their study show that "The low values of the bias error suggest that errors are close to Multiple-Doppler wind synthesis and should remain comprised between 1 and 1.5m/s" (see section 3.2, page 93).

These values are smaller than the radiosonde ones (between 1.8 and 2.52m/s). We added these values in section 4.2. "Bousquet et al. (2016) demonstrated that the bias error of RASTA wind data is comprised between 1 and 1.5 ms -1. In this study, it has been decided to use the same observation error as the one used for radiosondes, which increases with the altitude (from  $\approx$  1.8 ms -1 at 900 hPa to  $\approx$  2.52 ms -1 at 200 hPa)."

RASTA wind data have also been evaluated during the NAWDEX field campaign which occurred in Iceland (http://www.pa.op.dlr.de/nawdex/). In the following figures, RASTA wind retrieval were compared against radiosonde measurements. These Figures demonstrate that the observational error for RASTA wind data is of the same order of magnitude as that of radiosondes.

Bousquet, O., Delanoë, J. and Bielli, S. (2016), Evaluation of 3D wind observations inferred from the analysis of airborne and ground-based radars during HyMeX SOP-1. Q.J.R. Meteorol. Soc., 142: 86-94. doi:10.1002/qj.2710

3. I suggest that Figure 5, and explanations for Figures 4 and 5 will be modified. The increment in Fig. 5A is reflected the flight path of all observations, thus, all data points assimilated in this 3-h window should be presented in Fig. 5A. Moreover, the 1-h, 2-h, and 3-h assimilation window experiments include the observation until 0630, 0700, and 0730 UTC, respectively (L10 P7). I think that this different time limitations create the difference between panels in Figure 4 unlike the authors explanation on overpasses (L6-17 P8). Please exam and discuss this point of view.

Figure 5 (now Figure 6) has been modified: All the data that are assimilated in the RASTA\_3h experiment are now shown. The explanations have also been modified. Fig 5a is first described is section 5.2:

"Figure 5A represents the wind speed increments at approximately 4 km of altitude (model level 30) between the RASTA\_3h and the CTRL analysis. Wind directions are also indicated by the green (resp. black) arrows for the CTRL (resp. RASTA\_3h) analysis. The data points assimilated in the RASTA\_3h experiment until 07:30 UTC are also represented by the black data points.".

Then, Fig.5 B-D are explained at the beginning of section 5.2: *"Figure 5 (panels B to D) represents the wind speed differences of the RASTA\_3 h 1-, 2- and 3-h forecasts and the CTRL ones. At each forecast term, the black data points indicate the different RASTA locations which are available during a 1-h time window centred on the forecast time (forecast term \pm 30 minutes)."*

Reviewer 1 is correct, the different time limitations explain the differences in wind and direction in Figure 4 (now Figure 5). We added this explanation in the text.

4. It is amazing for me that RASTA\_3h in Figure 4 improved the wind filed even at the end of the assimilation window because the experiment did not employ FGAT. Since the RASTA data only exist in cloud area, 3 hours seems too long to assimilate the data appropriately. I understand that this is the motivation of the authors to conduct three experiments. If they use FGAT, the 3-h experiment may significantly improve the result. I recommend the authors to conduct the FGAT experiment additionally if possible.

Reviewer 1 is right, FGAT is a way to improve the handling of the time dimension in a 3D-Var scheme as it allows to compute the innovations (i.e. the observation-guess differences) at the time of the observations for different times during the assimilation window. For the AROME model, the FGAT option has been evaluated by Brousseau (2012) for moving platforms, but without any positive improvement in the subsequent forecasts (Brousseau et al. 2016, section 2). For observations from static platforms, the 3DVar without FGAT only uses the observations performed at the middle of the assimilation window. The FGAT option allows to estimate innovations for sub-hourly data from the same instrument at the same location. More observations are assimilated, but the 3D-Var minimisation, without time dimension, uses these several innovations at the middle of the assimilation window. This leads to an averaging and a smoothing effect on these observations and a loss of information on the temporal details, which is not desirable in a convective DA system. Therefore, in this study we decided to use conventional 3DVar to assimilate all the different kinds of observations in the same way.

Brousseau, 2012: Propagation of observed information into the AROME data assimilation and atmospheric model, PhD thesis, Université de Toulouse III – Paul Sabatier

Brousseau, P., Seity, Y., Ricard, D. and Léger, J. (2016), Improvement of the forecast of convective activity from the AROME-France system. Q.J.R. Meteorol. Soc., 142: 2231-2243. doi:10.1002/qj.2822

5. The authors used the median value of observations in a grid box (L16 P5) for thinning. If observational data distribute followed the Gaussian PDF and their number are large enough, the median and the mean values are the same. Usually "super observations" are made by the "mean" method in order to reduce representativeness errors and avoid noises. Therefore, the authors should explain why they adopt the "median" method instead of the mean.

The two approaches have been tested by the authors. After the data processing described in section 2.1 (whose description has been enhanced in the revised version of the paper), some spurious data were still occasionally present. Using a median filter, instead of the mean filter, helps to reduce the weight that these spurious observations can have when we calculate the "Super-observations". Besides, a median filter is also employed by Bousquet et al. (2016) and by Tabary et al. (2006) to calculate the "super observations" of ground-based radar Doppler velocity observations

Tabary, P., F. Guibert, L. Perier, and J. Parent-du-Chatelet, 2006: An Operational Triple-PRT Doppler Scheme for the French Radar Network. *J. Atmos. Oceanic Technol.*, **23**, 1645– 1656, https://doi.org/10.1175/JTECH1923.1

Bousquet, O., Delanoë, J. and Bielli, S. (2016), Evaluation of 3D wind observations inferred from the analysis of airborne and ground-based radars during HyMeX SOP-1. Q.J.R. Meteorol. Soc., 142: 86-94. doi:10.1002/qj.2710

6. English needs to be proofread by professional native speaker(s) with scientific background

The revised manuscript has been carefully copy-edited for English. Together with the copyediting standard service offered by Copernicus, we believe that the English should be sufficiently polished in the final version of our manuscript.

Responses to the minor comments:

L23 P1: "To fill the gap in clear air condition" I suggest the authors to refer the following articles, because wind observations in clear air can be also provided by Doppler lidars (air-born and ground-based), and clear air echoes (insects) by Doppler radars.

[Ground-based lidar] Kawabata, T., H. Iwai, H. Seko, Y. Shoji, K. Saito, S. Ishii, and K. Mizutani, 2014: Cloud-Resolving 4D-Var Assimilation of Doppler Wind Lidar Data on a Meso-Gamma-Scale Convective System. Mon. Wea. Rev., 142, 4484–4498, doi: 10.1175/MWR-D-13-00362.1.

[Air-born lidar] Weissmann, M., R. H. Langland, C.

Cardinali, P. M. Pauley, and S. Rahm, 2012: Influence of airborne Doppler wind lidar profiles near Typhoon Sinlaku on ECMWF and NOGAPS forecasts. Quart. J. Roy. Meteor. Soc., 138, 118–130, doi:10.1002/qj.896.

[Clear air echoes] Kawabata, T., H.Seko, K. Saito, T. Kuroda, K. Tamiya, T. Tsuyuki, Y. Honda, and Y. Wakazuki, 2007:An assimilation and forecasting experiment of the Nerima heavy rainfall with a cloud-resolving nonhydrostatic 4-dimensional variational data assimilation system. J. Meteor. Soc. Japan, 85, 255–276, doi:10.2151/jmsj.85.255.

The authors are grateful to Reviewer 1 for these references. We now refer to the suggested articles from L 23: "In clear air condition, wind observations can be provided by insectderived Doppler radar measurements (Kawabata et al., 2007; Rennie et al., 2011) or by Doppler lidars (Weissmann et al., 2012; Kawabata et al., 2014)." L19 P2: "has never been investigated" I did not understand what thing has never been investigated in the following "vertical profiles from Doppler W-band radar". "vertical profiles from Doppler radar"? "W-band radar"? "vertical profiles" by W-band radar? ("horizontal" winds have been done)? Please clarify.

We meant "vertical profiles by W-band radar". This has been rectified in the text (Doppler has been removed in the sentence).

L30 P2: "first" This is the same with the above. What is the first?

"First" means the assimilation of wind profiles measured by Doppler W-band radar. Since Doppler is redundant with "wind profiles", we removed Doppler.

L8 P3: "HyMeX-SOP1" What is this? Spell out it and add explanation.

HyMeX (Hydrological cycle in the Mediterranean Experiment) aims at a better understanding, quantification and modelling of the hydrological cycle in the Mediterranean, with emphasis on the predictability and evolution of extreme weather events (Drobinski et al., 2014). The HyMeX first Special Observing Period (HyMeX-SOP1, Ducrocq et al., 2014) took place during a 2-month period during the autumn 2012. The main goal of the HyMeX-SOP1 was to document the heavy rainfall and flashflood events which regularly affect the mediterranean area.

We added some informations L35, page 2 about the HyMeX-SOP1. For further explanations, the reader can refer to Ducrocq et al., (2014).

L28 P3: "six Cassegrain antennas" How do these six antennas observe three directions above and below the aircraft? Add explanation and, if possible, a schematic figure.

RASTA configuration during the HyMeX-SOP1 is given by Bousquet et al. (2016) (Figure 1). The radar is equipped with 6 antennas pointing either upward (antennas 1-3) or downward (antennas 4-6). Labels 1-6 refer to the 'upward transverse' (UT), 'zenith' (Z), 'upward backward' (UB), 'downward backward' (DB), 'nadir' (N) and 'downward transverse' (DT) antennas, respectively.

In the text, we added "A schematic figure of RASTA configuration during the HyMeX-SOP1 is given by Bousquet et al. (2016), their Figure 1". #1 (UT) #2 (Z)

---

## Author Response (AR2)

We are very grateful to the reviewers for their comments who significantly helped to improve this study. We corrected the minor corrections suggested to us by Reviewer 1. Our responses are given below in red. The correction in the article are also highlighted in red.

L33 P6: "the bias error"
Observation error means standard deviation, not bias. The authors should mention STD or RMSE instead of bias.

> Reviewer 1 is right; the bias is not important.
> We replaced *'Bousquet et al. (2016) demonstrated that the bias error of RASTA wind data is comprised between 1 and 1.5 ms $^{-1}$'* by*: 'Bousquet et al. (2016) demonstrated that the root-mean-square error of RASTA wind data versus ground-based centimetre-wavelength radars is on the order of 4 m s$^{-1}$. They argued that this error mainly originated from the sampling volume of ground-based radars being much larger than that of RASTA.'*

L28-20 P9: some of the most substantial … locations." I would like to draw the authors' attention to the lee side of the RASTA observations, where there are larger differences as longer the assimilation windows are (Fig. 6)

> We rectified the sentence: *"Besides, some of the most substantial differences are co-located with RASTA locations (black data points in Figure 6) and downstream of these locations."*

L33 P10: "113 mm) and … Therefore, the three …" The values 113 and 116 mm obtained from the 2h and 1h experiments are similar and/or larger than 114 mm in CTRL. Thus, only the 3h experiment is better.

> We rectified the sentence: *"The assimilation of RASTA wind data with smaller assimilation windows (2 and 3 hours) does not significantly impact the rainfall forecasts. Indeed, the maximum rainfall accumulation are of same order of magnitude in the RASTA$^{IOP7}$ (113 mm) and RASTA$^{IOP7}$ experiments (116 mm), compared to the CTRL$^{IOP7}$ one. "*

[Technical concerns]
L9 P1, L20 P13: "two hours and three hours" Remove the first "hours".

> We removed the first "hours" in both sentences.

L5 P10: "the wind speed" standard deviation of the wind speed?

> We rectified the sentence by "the standard deviation of the wind speed".

L23 P11, L13 P12: "0.5°both" "0.25°within" Insert single space after "°" (degree sign).

> We inserted the space after the degree sign.